# Economic impact and clinical benefits of clinical pharmacy interventions: A six-year multi-center study using an innovative medication management tool

**Watheq M. Alsetohy**[1☯]*, **Kareem A. El-fass**[2☯], **Seif El Hadidi**[3], **Mohammad F. Zaitoun**[4], **Osama Badary**[5], **Kareem A. Ali**[1], **Ahmed Ezz-Elden**[6], **Mohamed R. Ibrahim**[7], **Bahaa S. Makhlouf**[7], **Asmaa Hamdy**[8], **Noha S. El Baghdady**[9], **Maha Gamal Eldien**[10], **Sherif Allama**[11], **Amr A. Alashkar**[12], **Ahmed Seyam**[13], **Nanees A. Adel**[14], **Ahmed R. N. Ibrahim**[15], **Hany V. Zaki**[16,17]

**1** Medication Therapy Management and Pharmaceutical Services Sector, Headquarter, Cleopatra Hospitals Group–(CHG), Cairo, Egypt, **2** Syreon Middle East, Alexandria, Egypt, **3** University of Hertfordshire, New Administrative Capital, Cairo, Egypt, **4** Pharmaceutical Care Administration. Armed Forces Hospitals Southern Region, Khamis Mushayt, KSA, **5** Department of Clinical Pharmacy, Ain Shams University, Cairo, Egypt, **6** General Administration, Headquarter, Cleopatra Hospitals Group–(CHG), Cairo, Egypt, **7** Department of Pharmacy, Nile Badrawi Hospital, Cleopatra Hospitals Group, Cairo, Egypt, **8** High Institute of Public Health, Alexandria University, Alexandria, Egypt, **9** Clinical Pharmacy Practice Department, Faculty of Pharmacy, The British University in Egypt, El Sherouk City, Egypt, **10** Department of Pharmacy, Cleopatra Hospital, Cleopatra Hospitals Group, Cairo, Egypt, **11** Department of Pharmacy, Cairo Specialized Hospital, Cleopatra Hospitals Group, Cairo, Egypt, **12** Information Technology Sector, Headquarter, Cleopatra Hospitals Group–(CHG), Cairo, Egypt, **13** Department of Health Economics and Technology Assessment, Universal Health Insurance Authority (UHIA), Cairo, Egypt, **14** Cairo Specialized Hospital, Cleopatra Hospitals Group–(CHG), Cairo, Egypt, **15** Department of Clinical Pharmacy, College of Pharmacy, King Khalid University, Abha, Saudi Arabia, **16** Cleopatra Hospital, Cleopatra Hospitals Group–(CHG), Cairo, Egypt, **17** Department of Critical Care and Anesthesiology, Ain Shams University, Cairo, Egypt

☯ These authors contributed equally to this work.
* Watheq.setouhy@cleohc.com

**Data Availability Statement:** The data underlying the results presented in this study are available upon reasonable request from the Cleopatra

## Abstract

### Background

Increasing healthcare costs, particularly in Low- and Middle-Income Countries (LMICs) like Egypt, highlight the need for rational economic strategies. Clinical pharmacy interventions offer potential benefits by reducing drug therapy problems and associated costs, thereby supporting healthcare system sustainability.

### Objective

This study evaluates the economic impact and clinical benefits of clinical pharmacy interventions in four tertiary hospitals in Egypt by implementing an innovative tool for medication management, focusing on cost avoidance and return on investment (ROI), while accounting for case severity and drug therapy problem (DTP) resolution.

Hospitals Group. Due to confidentiality agreements, certain sensitive information, including service costs, tender details, and proprietary financial data, cannot be shared publicly. Access to these data will be granted only to researchers who meet the criteria for access to confidential information and have received the necessary approvals from the Cleopatra Hospitals Group's Institutional Data Access Committee. Data requests can be made via email to clinicarepro. app@cleohc.com.

**Funding:** This study was financially supported by the Deanship of Research and Graduate Studies at King Khalid University (https://dps.kku.edu.sa/en) through the Large Research Project grant (RGP2/ 391/45) awarded to ARNI. No additional external funding was received for this study. The funder had no role in study design, data collection and analysis, decision to publish, or preparation of the manuscript.

**Competing interests:** All authors declare that they have no conflict of interest. The authors confirm that the research was conducted in the absence of any personal, commercial, or financial relationships.

## Methods

Utilizing a digital tool, cost avoidance was assessed by considering the severity of patient cases and the effectiveness of clinical pharmacists' interventions in resolving identified DTPs. Additionally, ROI was calculated by incorporating both full-time equivalent (FTE) and non-FTE costs to evaluate the overall economic impact of clinical pharmacy services over a six-year period across four tertiary care hospitals in Egypt.

## Results

Over six years, a total of 492,612 patients were reviewed, leading to 19,240 comprehensive clinical pharmacy interventions. These interventions achieved an 88.63% resolution of DTPs, significantly reducing patient risk by preventing adverse DTP consequences, resulting in a total cost avoidance of EGP 265.32 million (USD 8.60 million) and an average ROI of 7.6 (760%). This underscores the substantial economic impact of clinical pharmacy services, particularly in LMICs and countries transitioning to universal health insurance coverage, where cost efficiency and patient safety are critical.

## Conclusion

This study underscores the importance of clinical pharmacy interventions in improving healthcare outcomes and generating significant economic benefits, particularly in low- and middle-income countries. By accounting for case severity and the level of DTP consequences, along with the efficiency of clinical pharmacist-led interventions in resolving DTPs, the economic impact of these services can be more accurately evaluated. These findings are essential for informing policy decisions, highlighting the critical role of clinical pharmacy services in supporting healthcare systems facing economic constraints.

## Introduction

Healthcare costs are rising worldwide, causing a rise in health expenditures to an extent that burdens healthcare systems and threatens the sustainability of services [1, 2]. In Egypt, pharmaceutical expenditure represents about 34% of total health expenditure, placing a heavy on healthcare system [3]. Drug Therapy Problems (DTPs) represent a substantial part of the health care expenditure; they compromise the clinical care of patients and its outcomes, especially for hospitalized ones [4–6]. DTPs prevalence in hospitalized patients ranges from 15.5 to 81% [7–9]. Half of them are potentially preventable [4]. Economically, patients with DTPs have twice the average hospitalization costs compared with those without [10]. Clinically, DTPs are implicated in 21% of hospital readmission cases, with evidence suggesting that 69% of these instances could potentially be preventable [11], DTPs contribute to prolonged hospital stays [12], and are associated with higher mortality rates [13], Addressing preventable DTPs is crucial for healthcare systems from both safety and cost perspectives.

Clinical pharmacists play a significant role in detecting and reducing DTPs [7], reducing costs, and optimizing patient outcomes. Clinical pharmacy interventions have been found to reduce DTPs in different medical specialties [14], The presence of Clinical Pharmacy has been reflected in cost savings to the healthcare system [15]. The cost saving has reached USD 2,657,820 per year and is expected to reach over USD 2200 million in Japan [7], In Spain, it

was estimated to be 64.30 Euros per patient per year, and the return on investment (ROI) would be 2.38 Euros [16]. In Arab countries, many studies have reported the positive clinical and economic impact of pharmaceutical interventions for DTPs [17, 18]. One of these studies, conducted in Egypt, reported a net economic benefit of more than USD 7.5 million per year [18].

Egypt is one of the low-middle-income countries (LMIC) that has been transforming its healthcare system to adopt universal health coverage (UHC) since 2018 [19, 20]. This transformation is faced with different challenges [3], particularly with the rising total healthcare expenditures, which have ranged from 4.36% to 5.16% of general government expenditure during the decade spanning 2010 to 2020 [19], In LMICs, the implementation of clinical pharmacy services is often challenged by limited resources and escalating healthcare costs. Despite these challenges, clinical pharmacy has been mandated in all Egyptian hospitals since 2013, and as part of hospital licensing since 2012 [21]. Different studies have reported significant contributions of clinical pharmacists in reducing DTPs [22, 23]. However, until now, there have been no published studies on the economic benefits of clinical pharmacy or the ROI of implementing clinical pharmacy services in Egyptian hospitals. Thus, the wider implementation of clinical pharmacy in private and public settings still faces the question of "Is it worthwhile?".

## Objectives

This study focuses on a longitudinal investigation of the dual advantages—economic and clinical—of clinical pharmacy interventions within a network of hospitals, utilizing a medication therapy management digital tool. The objective is to demonstrate the dual impact of clinically resolving DTPs while concurrently achieving substantial cost reductions by preventing adverse DTP consequences. This approach enables healthcare entities to offer more competitive value-based healthcare services by maintaining superior patient care and safety standards at a lower cost.

## Methods

### Study design and settings

A retrospective multicenter study to estimate the cost-avoidance resulting from clinical pharmacists' interventions across four tertiary care hospitals operated by Cleopatra Hospitals Group (CHG) in Egypt: Nile Badrawi, Cleopatra, Cairo Specialized, and Alshorouk Hospitals. The four settings are located in the largest two Egyptian governorates, Cairo and Giza, in terms of population [24]. These four settings collectively encompass 782 in-patient and 150 ICU beds, six cardiac catheter laboratories, 24 operation theatres, and 70 outpatient clinics. From January 2018 to October 2023, the hospital group served 5,189,012 patients across various specialties, including cardiology, hematology, and oncology, and performed 196,921 surgeries [25].

### Ethics consideration

The study protocol was reviewed and approved by the Institutional Review Board (IRB) of the British University in Egypt (Registration Code: CL-2404, approved on July 9, 2024).

### Data access and participant confidentiality

Data was accessed for research purposes on 21/07/2024, following CHG administration approval to access the internal dashboards for this study. The data was supplied in a manner

that ensured the authors did not have access to any information that could identify individual participants during or after data collection.

## Tool development

In 2017, the digital tool Clinicare Pro® Integrated Solutions was developed by Watheq M. Alsetohy to address DTPs and support medication management. Following consultations with experts from different healthcare disciplines, an in-depth literature review and technical feasibility analysis, this web-based application was integrated with the hospitals' enterprise resource planning (ERP) and hospital information system (HIS), Clinisys®, by December 2017. The application uses the Pharmaceutical Care Network Europe (PCNE) Classification for DTP definitions and incorporates an algorithm to categorize Medication Errors [6, 26], Economically, it initially implemented Lee et al. model [27], and later transitioned to adopt the Patanwala et al. [28] guidelines, which consider broader aspects and probabilities. The Clinicare Pro® app was developed using ASP. Net, C#, CSS, and JavaScript and is managed by the Azure Database for MySQL. The impact of Clinicare Pro® on DTP reporting and the acceptance of clinical pharmacist recommendations has been evaluated in a related study, demonstrating significant improvements in DTP detection and recommendation acceptance rates [29].

## Provision of clinical pharmacy services

Over 6 years, clinical pharmacists provided pharmaceutical care, starting each shift with patient reviews assigned via the CLINIcare Pro® App, based on Electronic medical record (EMR) data, and prioritized according to the guidelines of the Society of Hospital Pharmacists of Australia's guidelines [30]. Pharmacists selected review types (basic, intermediate, or advanced) for each patient following the PCNE statement on medication review [31]. DTPs were logged with recommendations and interventions, integrated with the HIS for cost tracking (see S1 and S2 Figs).

## Documentation of clinical interventions

Post-DTP identification, clinical pharmacists consulted with the responsible physicians, proposing interventions to manage the identified DTPs and assess potential harm avoidance. The acceptance of these interventions and the physicians' assessments of potential harm avoidance were documented. This documentation included evaluating the patient's expected trajectory without the interventions, using a risk classification framework and a consequence/probability matrix, in line with the Society of Hospital Pharmacists of Australia's guidelines [32]. The time for each intervention was estimated based on the duration from patient review to reporting, adjustable by pharmacists. Regular follow-ups were conducted to update the DTPs' status, noting resolutions or persistence.

## Patient days coverage

We measured the extent of clinical pharmacy service delivery using 'Patient Days Coverage', calculated as:

$$\text{Coverage} = \frac{\text{Patient days with clinical pharmacy services}}{\textit{Total patient days}} \times 100$$

Here, a 'patient day' represents the services provided to a patient over a 24-hour period [33]. This formula quantifies the proportion of patient days that included clinical pharmacy services.

## Cost avoidance

Cost-avoidance calculations for clinical pharmacy interventions were conducted using the CLINIcare Pro Solution® based on Patanwala et al. [28] This tool avoids counting duplicate interventions for the same patient. The ROI is calculated using CLINIcare Pro Solution® and Clinisys® (Version 2023.3.8), both integrated with Tableau Prep® (Version 2023.1) and Tableau® (Version 2023.1) for enhancing data consolidation and computational accuracy. Monetary conversions from the Egyptian Pound (EGP) to the United States Dollar (USD) are based on October 2023 rates from the Central Bank of Egypt (CBE) [34].

Cost avoidance was determined using the formula:

$$\text{Cost avoidance} = [\text{pTC} \times (\{\text{pCON} \times \text{cCON}\} + \text{DCS}) \times \text{iFactor} \times \text{sFactor}] - [\text{cPharm}]$$

Whereas the original formula by Patanwala et al. [28] considers only pTC = Probability of Trajectory Change, pCON = Probability of Consequence, cCON = Cost of Consequence, DCS = Direct Cost Savings, and cPharm = Cost of Pharmacy Arm, this study expanded the model to include new variables: iFactor = Clinical Intervention Implementation Factor, and sFactor = DTP Final Status Factor.

## Assuming probabilities

In this study, we focused on DTPs that could significantly alter patient trajectories if not addressed. The pTC parameter represents the probability that other healthcare professionals would identify and resolve the DTP if clinical pharmacists did not intervene. This value was adapted from Patanwala et al.'s quartiles (0, 0.25, 0.5, 0.75, 1) and was set to 1 in cases where comprehensive clinical pharmacy review was deemed essential—beyond what standard central pharmacy checks would uncover [28]. Adjustments to pTC were made only when the likelihood of intervention by other healthcare team members was considered plausible.

The pCON parameter represents the probability that an adverse consequence would occur if the clinical pharmacist did not intervene. When the probability of adverse consequences associated with unresolved DTPs was reported in the literature (e.g., the probability of stress ulcers in hospitalized patients without stress ulcer prophylaxis while clinically indicated) [35–38], the expert panel used these data as the basis for their assessments. In scenarios where such probabilities were not reported, the expert panel estimated the likelihood based on their clinical judgment. In both scenarios, the expert panel quantified the probability of incidence using the Nesbit probability scale, which standardizes probabilities into the following categories: A: Almost certain = 60%, B: Likely = 40%, C: Possible = 10%, D: Unlikely = 1%, E: Rare = 0% [32, 39].

**Cost of consequence (cCON).** cCON quantifies the cost associated with potential harm, calculated from the additional days a patient stays in the hospital due to harm (consequence days) and the per-day cost, using the following formula:

$$cCON = [\textit{Number of Consequence Days}] \times [\textit{Cost per Consequence Day}]$$

In this study, we will use two different methods to estimate the cCON, as follows:

**cCON.lit method:** the traditional approach, termed 'cCON.lit' in this study, assumes a fixed number of consequence days for all patients, typically two days, as proposed by many published studies [40–45], regardless of case severity or comorbidities. The 'Cost per Consequence Day' is calculated based on the unit bed-day cost, which is derived from the CHG cost center that specifies the cost of provided services and medical interventions, as listed in the price index for the fiscal year 2023 intended for payers.

**cCON.cleo panel method:** The study developed the cCON.cleo panel method in response to the significant adverse risks posed by DTPs in critically ill patients, attributed to factors

**Table 1. Estimation of 'Number of consequence days' using the cCON.cleo panel method.**

| Consequence Level[a] | Non-Critically Ill | Critically Ill |
|---|---|---|
| 1 Insignificant | 0 | 0 |
| 2 Minor | 0 | 0 |
| 3 Moderate | 0.75 days | 1.5 days |
| 4 Major | 1.5 days | 3.5 days |
| 5 Catastrophic | 2.25 days | 5 days |

[a] Consequence Level refers to the impact of the Drug Therapy Problem (DTP) on the patient, assessed under the assumption that no clinical interventions are undertaken.

such as unstable health status and complex medication regimens [46, 47], coupled with the diverse range of potential consequences of DTPs [32]. This approach was shaped by a multidisciplinary panel of consultants with expertise in neurology, cardiology, intensive care, internal medicine, and clinical pharmacy, boasting experiences ranging from 10 to over 26 years. Leveraging their longitudinal experiences with the consequences of DTPs, alongside CHG data on the average length of stay following DTP incidents and existing literature indicating that DTPs can extend hospitalization by 1.7 to 4.6 days [48–51], the panel reached a consensus on the necessary extensions of hospital stay for DTPs at various consequence levels, as detailed in Table 1. Additional details on the panel are provided in S1 File.

**Direct cost savings (DCS).** DCS represents the direct cost savings attributable to the intervention. It is calculated by comparing costs before and after clinical pharmacy intervention, covering several aspects such as medication additions or discontinuations, modifications, and other services including extra laboratory monitoring requested due to clinical pharmacists' interventions (e.g. Therapeutic Drug Monitoring (TDM). For detailed DCS calculations, for detailed DCS calculation, see S2 File.

**Implementation (iFactor) and problem status (sFactor).** This study introduces novel variables to the Patanwala et al. model [28], the Implementation Factor (iFactor) and the Problem Status Factor (sFactor), both estimated by CLINIcare pro®. The iFactor evaluates the implementation of clinical pharmacy interventions by physicians, assigning values of 0 for non-implementation, 0.4129 for uncertain implementations (reflecting 50% of the acceptance rate of interventions with known acceptance), and 1 for full implementation. The sFactor assesses DTP resolution, with 0 for unresolved issues, 0.471 for unknown resolutions (reflecting 50% of the DTPs resolution rate for accepted interventions), 0.5 for partially resolved, and 1 for fully resolved DTPs. This approach has been developed to account for scenarios where the acceptance of interventions is uncertain or where drug therapy problems remain unresolved.

**Cost of the pharmacist arm (cPharm).** cPharm represents the exact full-time equivalent (FTE) cost associated with the time taken by the clinical pharmacist to identify the DTP and recommend an effective intervention to solve it. It is calculated using the formula:

$$\text{cPharm} = [\text{Intervention time in minutes}] \times [\text{FTE cost per minute}]$$

For an applied example of Cost-Avoidance Calculations, see S3 File.

**Return on investment (ROI).** ROI, a metric commonly used in business and investment sectors to evaluate investment effectiveness, is less explored in evaluating the economic outcomes of health interventions [52]. This study employs ROI as a framework for converting costs and benefits in clinical pharmacy services into ROI, using data from the Cleopatra Hospitals' Clinisys® (HIS, ERP) and CLINIcare pro® systems. It is important to note that

calculated ROI are exclusive to patient-specific clinical pharmacists' interventions and excludes the ROI from other activities at the organizational level (e.g., formulary management). This distinction ensures a focused analysis of the direct impact of clinical pharmacists' interventions on economic outcomes. The ROI for clinical pharmacy services is calculated as follows:

$$\text{ROI} = \frac{\sum[\text{PTC} \times (\{\text{pCON} \times \text{cCON}\} + \text{DCS}) \times \text{iFactor} \times \text{sFactor}] - \sum[\text{Cost of Investment}]}{\sum[\text{Cost of Investment}]}$$

Investment cost includes clinical pharmacy FTE costs and expenses (salaries, compensations, incentives, and profit sharing) [53], and non-FTE-linked costs (training, programs, medical wear, software, and overhead expenses). This methodology adheres to financial principles [54], allowing the estimation of returns per 1 EGP or USD invested in clinical pharmacy services.

### Data analysis

Clinical pharmacy interventions from the CHG four hospitals were stored in an SQL database, integrated with Tableau Prep Builder® and Tableau® (Version 2023.1) for advanced data analysis. The dashboard created from these data provides insights into patient coverage, medication error severity, and DTPs using the PCNE classification. It identifies prevalent drug therapy issues, their causes, untreated conditions, intervention acceptance rates, and DTP resolution rates.

Economically, the dashboard offers a breakdown of the cost avoidance from interventions across specialties, medications, and therapeutic groups. It also includes a comprehensive ROI analysis for the patient-specific clinical pharmacy services, segmented by specialties and case severity. For more details, refer to S4 File.

## Results

### Patient days coverage and clinical interventions

During the study period from January 2018 to October 2023, clinical pharmacists achieved a patient-days coverage of 71.56% for 492,612 patients (n = 492,612) across four hospitals. Full coverage (100%) was provided to 113,298 patients in emergency and critical care areas, whereas 66.11% coverage was extended to 379,314 patients in less critical areas. Additionally, they identified and reported 19,240 interventions that were considered to potentially alter patients' treatment trajectories.

### Cost avoidance

The clinical pharmacy interventions resulted in a total cost avoidance of EGP 265.32 million (USD 8.60 million). The cost avoidance for critically ill patients amounted to EGP 251.36 million (USD 8.15 million), compared with EGP 13.96 million (USD 0.45 million) for non-critically ill patients.

The average cost avoidance per clinical pharmacist intervention amounted to EGP 13,790 (USD 447). In terms of case severity, this average was higher for critically ill patients at EGP 20,003 (USD 659) and lower for non-critically ill patients at EGP 2,091 (USD 68) (Table 2).

### Impact of the case severity and consequence level scenario

For critically ill patients, the cCON.cleo panel scenario yielded EGP 251.36 million (USD 8.15 million), representing a 50.86% increase compared with the cCON.lit scenario, which

**Table 2. Summary of clinical pharmacy intervention outcomes and economic benefits.**

| Measured Value | Non-Critically Ill | Critically Ill | Grand Total |
|---|---|---|---|
| Number of Interventions (NI) | 6,670 | 12,570 | 19,240 |
| Number of reviewed patients | 379,314 | 113,298 | 492,612 |
| Clinical Pharmacy Coverage* | 66.11% | 100% | 71.56% |
| Cost Avoidance † | EGP 13.96 million (USD 0.45 million) | EGP 251.36 million (USD 8.15 million) | EGP 265.32 million (USD 8.60 million) |
| Cost Avoidance cCON.lit Scenario ‡ | EGP 29.41 million (USD 0.95 million) | EGP 166.62 million (USD 5.40 million) | EGP 196.03 million (USD 6.36 million) |
| Cleo Panel Scenario Impact on cost avoidance § | -52.55% | +50.86% | +35.35% |
| AVG. Cost Avoidance/Intervention | EGP 2,091 (USD 68) | EGP 20,003 (USD 659) | EGP 13,790 (USD 447) |
| % Acceptance | 75.92% | 85.41% | 82.12% |
| % DTP Resolution ‖ | 89.65% | 88.15% | 88.63% |

AVG, Average; DTPs, Drug therapy problems.

* The percentage of total patient days in the hospital covered by clinical pharmacists for 492,612 reviewed patients.

† Cost avoidance is based on case severity and DTP consequence levels, presuming the absence of clinical intervention (cCON.cleo panel).

‡ Cost avoidance is calculated assuming a fixed number of consequence days for all cases, regardless of severity or DTP consequence levels, assuming clinical interventions are not applied. (cCON.lit scenario).

§ The percentage change in total cost avoidance between the cCON.lit method and the cCON.cleo panel method.

‖ The Rate of DTPs resolution for accepted clinical pharmacy interventions.

amounted to EGP 166.62 million (USD 5.40 million). In contrast, for non-critically ill patients, the cCON.cleo panel scenario resulted in EGP 13.96 million (USD 0.45 million), which was 52.55% lower than the cCON.lit-based scenario, totalling EGP 29.41 million (USD 0.95 million) (Table 2).

## Return on investment (ROI)

The study revealed yearly variations in ROI, with the highest at 9.3 (930%) in 2019 and the lowest at 6.5 (650%) in 2018. Yearly values were 6.5, 9.3, 9.0, 6.6, 6.8, and 7.1 from 2018 to 2023, respectively, as illustrated in Fig 1. The cumulative ROI for the 6-year period was 7.6 (760%).

## Acceptance and DTPs resolution rate

The overall acceptance rate for clinical pharmacist interventions was 82.12%, which was higher for critically ill patients (85.41%) than for non-critically ill patients (75.92%). The resolution rate of DTPs for accepted interventions was 88.63% overall, with 88.15% for critically ill patients and 89.65% for non-critically ill patients (Table 2).

## Causes of drug therapy problems (DTPs)

Analysis of the identified DTPs showed 'Condition without Treatment' as the most common (26.2% of DTPs), resulting in EGP 76.30 million (USD 2.47 million) cost avoidance. The predominant untreated conditions were cardiovascular (39%) and metabolic disorders (22%). In contrast, 'Dose timing instructions wrong, unclear, or missing' accounted for only 1.3% of DTPs but had the highest acceptance rate at 90.8%, resulting in EGP 4.04 million (USD 0.13 million) cost avoidance. In terms of DTP resolution, 'Inappropriate duplication of a therapeutic group or active ingredient' had the highest rate at 92.5%, while 'Duration of treatment too long' had the lowest at 84.2%, below the 88.63% average (Table 3).

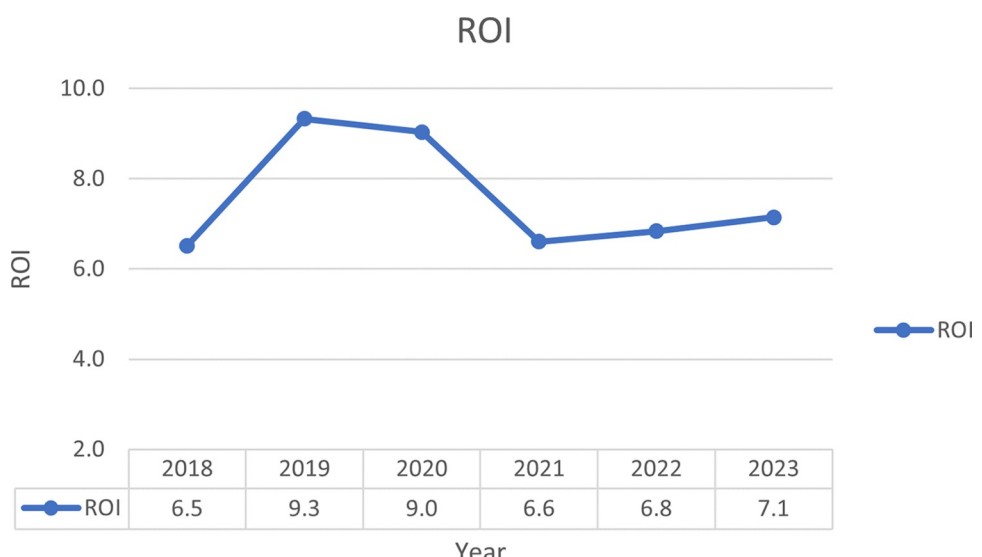

**Fig 1. A time series for the Return on Investment (ROI) of clinical pharmacy services during the study period.**

**Table 3. Causes of drug therapy problems (DTPs).**

| Root Cause | Cost avoidance | Percentage | % Acceptance | % DTPs Resolution |
|---|---|---|---|---|
| Condition without Treatment | EGP 76.30 million (USD 2.47 million) | 26.2% | 89.4% | 90.8% |
| No indication of drug | EGP 60.93 million (USD 1.98 million) | 23.8% | 70.4% | 89.0% |
| Inappropriate drug according to guidelines/formulary | EGP 34.22 million (USD 1.11 million) | 11.8% | 83.0% | 84.3% |
| Dose too high | EGP 21.96 million (USD 0.71 million) | 8.2% | 86.2% | 86.6% |
| Dose too low | EGP 20.60 million (USD 0.67 million) | 7.5% | 85.6% | 88.3% |
| Dosage regimen too frequent | EGP 6.33 million (USD 0.21 million) | 3.3% | 82.6% | 87.5% |
| Duration of treatment too long | EGP 7.88 million (USD 0.26 million) | 3.1% | 81.4% | 84.2% |
| No or inappropriate outcome monitoring (incl. TDM) | EGP 4.48 million (USD 0.15 million) | 2.9% | 82.6% | 91.5% |
| Inappropriate form/formulation | EGP 7.99 million (USD 0.26 million) | 2.9% | 78.5% | 89.9% |
| Dosage regimen not frequent enough | EGP 4.72 million (USD 0.15 million) | 2.7% | 82.5% | 89.3% |
| Inappropriate combination | EGP 6.68 million (USD 0.22 million) | 2.4% | 83.0% | 87.7% |
| Inappropriate duplication of therapeutic group | EGP 4.30 million (USD 0.14 million) | 2.1% | 82.8% | 92.5% |
| Dose timing instructions wrong, unclear or missing | EGP 4.04 million (USD 0.13 million) | 1.3% | 90.8% | 87.2% |

TDM = Therapeutic Drug Monitoring

## Interventions for drug therapy problems (DTPs)

The analysis of clinical pharmacy interventions revealed that 40.7% resulted in holding inappropriate medication, yielding EGP 112.15 million (USD 3.64 million) in cost avoidance. 'Instructions for use Added/changed' interventions, at 0.3%, resulted in EGP 1.29 million (USD 0.04 million) and had the highest acceptance rate of 92.9%, above the 82.12% average. In contrast, 'Drug paused or stopped' interventions had a lower acceptance rate at 76%. For DTP resolution, 'New Lab service requested' had the highest rate at 92.2%, while 'Drug Modified to New drug, Dose, Duration, or Form' had the lowest rate at 87.3%, both compared with the 88.63% average (Table 4).

## Therapeutic groups

DTPs were categorized by the main therapeutic group of the anatomical therapeutic chemical code (ATC), with 'Antibacterial drugs' leading at 30.5%, resulting in a cost-avoidance of EGP 63.61 million (USD 2.06 million). 'Vitamins and Minerals' interventions had the highest acceptance rate at 90%, while 'Antibacterial drugs' had a lower acceptance rate at 75.4%. 'Drugs for functional gastrointestinal disorders' achieved the highest resolution rate at 93.2%, while 'Antibacterial drugs' had the lowest resolution rate at 85.8% (Table 5).

## Discussion

In this study, we primarily focused on the economic extent to which interventions by clinical pharmacists effectively addressed the challenge of DTPs and contributed to clinical benefits. The findings demonstrate that these interventions clinically resolved 88.63% of the total identified DTPs.

Economically, the clinical pharmacy interventions resulted in a significant cost avoidance of EGP 265.32 million (USD 8.60 million), with an average cost-avoidance of EGP 13,790 (USD 447) per intervention, showing a notable disparity between critically ill patients (EGP 20,003 or USD 659) and non-critically ill patients (EGP 2,091 or USD 68). These findings highlight the substantial financial impact of clinical pharmacy interventions, not only in terms of direct savings but also in terms of cost avoidance, contributing significantly to economic efficiency and fiscal sustainability in Egypt's current economic status and budget constraints.

The current methodology for estimating cost avoidance includes new factors, considering the implementation of interventions and their efficiency in resolving DTPs, and additionally accounts for case severity and the level of consequences associated with DTPs, to better reflect

**Table 4. Clinical pharmacy interventions for drug therapy problems (DTPs).**

| Intervention (group) | Cost avoidance | Percentage | % Acceptance | % DTPs Resolution |
|---|---|---|---|---|
| Drug paused or stopped | EGP 112.15 million (USD 3.64 million) | 40.7% | 76% | 87.8% |
| Drug Modified to a New Drug, Dose, Duration, or Form | EGP 65.24 million (USD 2.12 million) | 26.8% | 83.4% | 87.3% |
| New drug added | EGP 67.80 million (USD 2.20 million) | 24.1% | 89.1% | 90.1% |
| New Lab, service requested † | EGP 18.83 million (USD 0.61 million) | 8.2% | 87.1% | 92.2% |
| Instructions for use Added/changed | EGP 1.29 million (USD 0.04 million) | 0.3% | 92.9% | 91.2% |

† For Therapeutic Drug Monitoring (TDM) and other recommended non-drug therapy interventions.

**Table 5. Main therapeutic categories of drug therapy problems (DTPs).**

| Therapeutic Category | Cost avoidance | Percentage | % Acceptance | % DTPs Resolution |
|---|---|---|---|---|
| Antibacterial drugs † | EGP 63.61 million (USD 2.06 million) | 30.5% | 75.4% | 85.8% |
| Antithrombotic agents | EGP 28.55 million (USD 0.93 million) | 10.1% | 81.8% | 90.3% |
| Blood substitutes, perfusion solutions, and Total Parenteral Nutrition | EGP 25.30 million (USD 0.82 million) | 9.2% | 87.1% | 88.2% |
| Antihypertensive drugs | EGP 21.06 million (USD 0.68 million) | 8.0% | 84.4% | 87.5% |
| Vitamins and Minerals | EGP 20.93 million (USD 0.68 million) | 7.2% | 90% | 90.9% |
| Drugs for acid-related disorders | EGP 10.84 million (USD 0.35 million) | 5.4% | 76.9% | 91.8% |
| Lipid modifying agents | EGP 6.81 million (USD 0.22 million) | 2.4% | 85.5% | 89.1% |
| Drugs used in diabetes | EGP 4.86 million (USD 0.16 million) | 2.0% | 86.1% | 88.4% |
| Drugs for functional gastrointestinal disorders | EGP 4.63 million (USD 0.15 million) | 1.6% | 89.8% | 93.2% |
| Antigout preparations | EGP 4.98 million (USD 0.16 million) | 1.6% | 85.9% | 92.7% |

† This category includes only antimicrobial agents and does not encompass other types of anti-infective agents.

clinical practice. This approach resulted in an average annual cost avoidance of EGP 45,483,275.17 (USD 1,474,747.42). Compared with similar studies in the MENA region, such as those by Al-Maqbali et al. [55] in Oman and Abushanab et al. [56] in Qatar, which reported annual cost avoidance of USD 438,931 and USD 621,106, respectively, our findings showed higher cost avoidance. These studies were generally shorter, retrospective, and their results were projected for one year. Similarly, Tasaka et al. [7] in Japan reported an annual cost avoidance of USD 1,328,910 from clinical pharmacy interventions, focusing primarily on drug interventions and their correlation with the risk of adverse drug reactions (ADRs). In contrast, this study included both drug and non-drug interventions and dynamically adjusted costs based on patient case severity and the risk of adverse DTP consequences.

Furthermore, the 'cCON.cleo panel' approach showed an overall 35.35% increase in cost avoidance compared to the 'cCON.lit' literature approach [40–45], with a significant 50.86% increase among critically ill patients and a 52.55% decrease in non-critically ill patients. This overall increase is largely due to the difference in the level of adverse consequences between critically ill and non-critically ill patients, alongside the significantly higher cost of patient care in critical care compared to non-critical care settings [57]. These findings emphasize the need for tailored economic models that account for patient severity and specific clinical circumstances. Moreover, they align with the CHG strategy of prioritizing clinical pharmacy services for critically ill patients, who face a higher risk of adverse DTP consequences, as supported by various studies and practice guidelines [30, 46, 58].

Additionally, the study introduced a method for estimating ROI, revealing a net return of 7.6 (760%) on the cumulative investment in clinical pharmacy services over six years, supporting the case for continued funding and expansion of these services. In comparison, other studies have used the benefit-cost ratio (BCR) as a metric. For instance, Perez et al. [59], Jourdan et al. [60], and Chan et al. [61] reported BCRs of 4.81, 5.09, and 3.99 USD for each USD invested in clinical pharmacy services. These studies typically included only FTE costs and

expenses, whereas our analysis expanded the scope to include all service-related costs, covering both FTE and non-FTE categories.

Clinically, clinical pharmacists provided services to 492,612 patients, achieving 71.56% coverage of their patient days, with 100% coverage for critically ill patients and 66.11% coverage for non-critically ill patients' patient days. Medication appropriateness was reviewed in accordance with the Joint Commission International Accreditation Standards for Hospitals (JCI), and the Egyptian General Authority for Health Care Accreditation & Regulation (GAHAR) [62, 63]. The most common cause of DTPs was 'Condition without Treatment' (26.2%), leading to costs of EGP 76.30 million (USD 2.47 million). This informed the development and implementation of clinical decision-support actions and protocols which warrants further clinical and economic evaluation in future studies. The second most common causes of DTPs were identified as 'No Indication for a Drug' (23.8%) and 'Inappropriate Drug Use According to Guidelines/Formulary' (11.8%). These causes were particularly prevalent with antibacterial drugs, which emerged as the main therapeutic category of DTPs, accounting for 33.5%. Typically, these types of DTPs were resolved mainly by discontinuing the inappropriate medication. This approach was confirmed by our findings, which showed that 47.7% of clinical pharmacist interventions involved pausing or stopping drugs. Recognizing this early on, the authors and hospital pharmacy teams established an Antimicrobial Stewardship Program (ASP) in collaboration with medical teams. This ASP, primarily informed by real-time data from the DTPs dashboard and local antibiogram for each hospital, led to the development of protocols and significant clinical and economic impacts, as reported in a related study [29]. Furthermore, many studies have highlighted drug dosing as a primary cause of DTPs [7, 60, 64]. In our study, this issue accounted for 15.7% of the total causes and was primarily managed by hospital pharmacists. Clinical pharmacists intervened in more controversial and debatable dosing problems, indicating a specialized focus within the scope of clinical pharmacy practice.

While many studies have focused solely on analyzing accepted interventions [44, 45], the current study reported a notable 82.12% acceptance rate, reflecting the strong collaboration between clinical pharmacists and other healthcare professionals for enhancing patient care quality. Also, the acceptance rate was subject to further analysis to identify the causes of rejection for certain clinical interventions and to pinpoint areas for improvement.

Some limitations should be acknowledged in this study. First, as it was conducted retrospectively in Egypt's largest private hospitals, the results may not be fully generalizable to other settings, such as governmental and teaching hospitals, where different challenges and cost structures, such as free services, might apply. However, it is important to recognize the significant role of the private sector in Egypt's healthcare sector, where 60% of healthcare expenses are out-of-pocket (OOP) [3]. Additionally, with the implementation of the UHC and the segregation between healthcare financing and service provision, all public and private healthcare facilities can be contracted by the Universal Health Insurance Authority (UHIA) [20]. This evolving framework could enhance the relevance of findings from private settings to the broader Egyptian healthcare system, including governmental institutions.

While incorporating a matched cohort or comparing data from hospitals without clinical pharmacy services would have strengthened the conclusions, this was not feasible due to logistical and ethical considerations in the study context. Instead, the focus was on evaluating the pharmacoeconomic impact of clinical pharmacy interventions and assessing the clinical benefits at the level of DTPs resolution. Additionally, the significant number of participants and the longitudinal design of the study strengthen the results and enhance their external validity, particularly in contexts similar to those of LMICs. Although expert panel assessments provide valuable insights based on clinical experience, they may introduce subjectivity. Many studies have used expert panel assessments in cost-avoidance models; however, we enhanced this

approach by including experts from diverse clinical disciplines, facilitating structured discussions to achieve consensus, and framing the assessments within a structured consequence/probability matrix. This framework ensures consistency and reduces potential bias in the probability estimates [32, 42].

Of note, the study did not involve the documentation and outcome evaluation of the non-patient-specific clinical pharmacy services (e.g., formulary management). Furthermore, its analysis did not encompass basic yet essential pharmacy interventions, including medication reconciliation interventions, simple dose adjustments, and medication discontinuation recommended by the inpatient pharmacy team. The study focused only on the interventions of the clinical pharmacy staff with postgraduate degrees or structured training. Such focus could result in undercalculating the actual number of pharmacy interventions and their impact on DTPs resolution and ROI. However, incorporating these non-patient-specific activities and multidisciplinary efforts could be the focus of future studies with a broader scope to encompass all clinical pharmacy services. Such studies would require the development of cost-avoidance and ROI models that incorporate additional variables and economic factors associated with these activities.

## Conclusion

The study underscores both the economic and clinical benefits of clinical pharmacy interventions, demonstrated through significant cost avoidance and a high-resolution rate of DTPs. These findings highlight the potential value of integrating clinical pharmacy services and related digital tools more broadly, particularly in LMIC settings. Moreover, including case severity and the consequences level of DTPs in economic evaluations emphasizes the need for continued research to explore their comprehensive implications on healthcare systems.

## Supporting information

**S1 Fig. Detailed screenshots from CLINIcare Pro® app illustrating the user interface used by clinical pharmacists for patient reviews and intervention logging.**
(TIF)

**S2 Fig. Detailed screenshots from CLINIcare Pro® app illustrating the user interface used by clinical pharmacists for documenting interventions based on the Pharmaceutical Care Network Europe (PCNE) classification.**
(TIF)

**S1 File. Multidisciplinary panel composition and expertise, detailing consultants' roles in reaching consensus on hospital stay extensions for drug therapy problems (DTPs).**
(PDF)

**S2 File. Comprehensive breakdown of the direct cost savings (DCS) calculations derived from clinical pharmacy interventions.**
(PDF)

**S3 File. Example of cost-avoidance analysis illustrating the step-by-step process used to calculate cost-avoidance for a specific clinical pharmacy intervention.**
(PDF)

**S4 File. Visual presentation from the drug therapy problems (DTPs) Dashboard, including multiple figures showing the distribution, types, resolutions of DTPs, and associated cost avoidance.**
(PDF)

## Acknowledgments

We express our deep appreciation to the CHG Big Data Management team, led by Aly Mahmoud, for their exceptional efforts. The authors also extend their deepest gratitude to the clinical pharmacy teams at Nile Badrawi Hospital, Cleopatra Hospital, Cairo Specialized Hospital, and Alshorouq Hospital, as well as the administrative support from Cleopatra Hospitals Group, for their crucial contributions to this research. We sincerely thank Yasmin Kamel, Muhamed Rafaat, Mai E. Afifi, Galal A. Abou El-Ella, Omar A. Tageldin, Sara T. Nassar, Mahmoud ElAlfy, and Noura M. Mohamed for their invaluable expertise and dedication. Their collective efforts were instrumental in the successful completion of this study.

## Author Contributions

**Conceptualization:** Watheq M. Alsetohy, Kareem A. El-fass.

**Data curation:** Watheq M. Alsetohy, Kareem A. Ali, Mohamed R. Ibrahim, Bahaa S. Makhlouf, Maha Gamal Eldien, Sherif Allama, Ahmed Seyam.

**Formal analysis:** Watheq M. Alsetohy.

**Funding acquisition:** Ahmed R. N. Ibrahim.

**Investigation:** Watheq M. Alsetohy, Seif El Hadidi, Mohammad F. Zaitoun, Osama Badary, Kareem A. Ali, Ahmed Ezz-Elden, Mohamed R. Ibrahim, Bahaa S. Makhlouf, Asmaa Hamdy, Noha S. El Baghdady, Maha Gamal Eldien, Sherif Allama, Ahmed Seyam, Nanees A. Adel, Ahmed R. N. Ibrahim, Hany V. Zaki.

**Methodology:** Watheq M. Alsetohy, Kareem A. El-fass.

**Project administration:** Watheq M. Alsetohy.

**Resources:** Ahmed Ezz-Elden, Nanees A. Adel, Hany V. Zaki.

**Software:** Watheq M. Alsetohy, Amr A. Alashkar.

**Supervision:** Watheq M. Alsetohy, Ahmed Ezz-Elden, Nanees A. Adel, Hany V. Zaki.

**Validation:** Kareem A. El-fass, Seif El Hadidi, Mohammad F. Zaitoun, Osama Badary, Asmaa Hamdy, Noha S. El Baghdady, Ahmed R. N. Ibrahim.

**Visualization:** Watheq M. Alsetohy, Kareem A. El-fass.

**Writing – original draft:** Watheq M. Alsetohy, Seif El Hadidi, Mohammad F. Zaitoun.

**Writing – review & editing:** Kareem A. El-fass, Seif El Hadidi, Mohammad F. Zaitoun, Osama Badary, Ahmed Ezz-Elden, Asmaa Hamdy, Noha S. El Baghdady, Ahmed Seyam, Nanees A. Adel, Ahmed R. N. Ibrahim, Hany V. Zaki.

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
