## [Decision Letter · Decision Letter 0]

18 Nov 2024

PONE-D-24-37096Economic Impact and Clinical Benefits of Clinical Pharmacy Interventions: a six-year multi-center study Using an Innovative Medication Management Tool.PLOS ONE

Dear Dr. Alsetohy,

Thank you for submitting your manuscript to PLOS ONE. After careful consideration, we feel that it has merit but does not fully meet PLOS ONE’s publication criteria as it currently stands. Therefore, we invite you to submit a revised version of the manuscript that addresses the points raised during the review process.

We look forward to receiving your revised manuscript.

Kind regards,

Firomsa Bekele, Msc

Academic Editor

PLOS ONE

**Journal Requirements:**

2. In the online submission form, you indicated that your data is available only on request from a third party. Please note that your Data Availability Statement is currently missing the contact details for the third party, such as an email address or a link to where data requests can be made. Please update your statement with the missing information. 

Reviewers' comments:

Reviewer's Responses to Questions

**Comments to the Author**

1. Is the manuscript technically sound, and do the data support the conclusions?

Reviewer #1: Yes

2. Has the statistical analysis been performed appropriately and rigorously? 

Reviewer #1: Yes

3. Have the authors made all data underlying the findings in their manuscript fully available?

Reviewer #1: No

4. Is the manuscript presented in an intelligible fashion and written in standard English?

Reviewer #1: Yes

5. Review Comments to the Author

**Reviewer #1:** Cumulative ROI is represented but an yearly analysis could derive more on the trends as per yearly basis.

Incorporating a matched cohort or comparing data from hospitals without clinical pharmacy services could strengthen the conclusions. If a control group is not possible, stating this constraint more explicitly might be beneficial.

While the probability estimates used in cost-avoidance calculations are plausible, further information on their derivation and validation would improve the manuscript's clarity. It would be particularly useful to address any potential limitations of expert panel assessments in establishing probabilities.

Consider explaining the ramifications of removing non-patient-specific interventions, as this may underestimate the true benefit of clinical pharmacy services.

6. PLOS authors have the option to publish the peer review history of their article (what does this mean?). If published, this will include your full peer review and any attached files.

Reviewer #1: No

---

## [Author Response · Author response to Decision Letter 0]

26 Nov 2024

Point – to – Point reply to reviewers’ comments.

We appreciate the valuable feedback provided by Reviewer #1, which has significantly contributed to enhancing the quality and clarity of our manuscript. Below, we address the comments point by point:

1. Cumulative ROI is represented but a yearly analysis could derive more on the trends as per yearly basis.

Thank you for the suggestion to include ROI trends on a yearly basis. While the current results section highlights the highest and lowest ROI values during the study period, we acknowledge that presenting a detailed yearly breakdown can provide additional insights. The yearly ROI trends are already depicted in Fig. 1, showcasing variations across the six years. We have updated the ROI results section to include the yearly ROI breakdown, and to enhance clarity, we explicitly mentioned that Fig. 1 provides a detailed depiction of the annual trends. This will ensure that readers can readily access and interpret the yearly variations in ROI.

2. Incorporating a matched cohort or comparing data from hospitals without clinical pharmacy services could strengthen the conclusions. If a control group is not possible, stating this constraint more explicitly might be beneficial.

Thank you for the suggestion to incorporate a matched cohort or comparison with data from hospitals without clinical pharmacy services to strengthen the conclusions. In the revised version, we have explicitly highlighted that including a control group was not feasible due to logistical challenges, such as the lack of standardized data from comparable hospitals, and ethical considerations, including the withholding of clinical pharmacy services that are integral to patient care. We have also clearly stated this constraint in the updated limitations section, ensuring transparency and aligning with your recommendation. The study, therefore, focuses on evaluating the pharmacoeconomic impact of clinical pharmacy interventions and assessing clinical benefits at the level of DTPs resolution. We appreciate your valuable feedback and have addressed this thoroughly in the revised manuscript.

3. While the probability estimates used in cost-avoidance calculations are plausible, further information on their derivation and validation would improve the manuscript's clarity. It would be particularly useful to address any potential limitations of expert panel assessments in establishing probabilities.

Thank you for your comment regarding the derivation and validation of the probability estimates used in the cost-avoidance calculations. The manuscript has been updated to clarify how the expert panel quantified and calculated the probabilities, including the use of the Nesbit probability scale for standardization. Additionally, the potential limitation of expert panel subjectivity has been explicitly addressed in the limitations section to ensure transparency.

4. Consider explaining the ramifications of removing non-patient-specific interventions, as this may underestimate the true benefit of clinical pharmacy services.

Thank you for your thoughtful comment regarding the exclusion of non-patient-specific activities. This study focuses on clinical pharmacist-led interventions, with the cost estimation model tailored to evaluate their unique contributions. The pTC (probability of trajectory change) parameter ensures the assessment remains specific to the direct impact of clinical pharmacists. We recognize that non-patient-specific activities, such as formulary management, ASP, and IV admixture services, are valuable components of clinical pharmacy services. However, these often involve multidisciplinary teams and were beyond the scope of this study. To address this, we have clarified in the limitations section that incorporating these activities could be explored in future studies with a broader scope, which would require a comprehensive cost-avoidance and ROI models to account for additional variables and economic factors. This study provides a precise evaluation of clinical pharmacist-led interventions and lays the groundwork for future research on the broader contributions of clinical pharmacy services.

We hope these revisions and clarifications address your concerns adequately. We are grateful for your insightful suggestions, which have helped improve the manuscript's scientific rigor and applicability. Please let us know if further adjustments are required.

Warm regards,

Watheq M. Al Setohy

---

## [Editor Report · Decision Letter 1]

29 Dec 2024

Economic impact and clinical benefits of clinical pharmacy interventions: a six-year multi-center study using an innovative medication management tool.

PONE-D-24-37096R1

Dear Dr. Alsetohy,

We’re pleased to inform you that your manuscript has been judged scientifically suitable for publication and will be formally accepted for publication once it meets all outstanding technical requirements.

Kind regards,

Firomsa Bekele

Academic Editor

PLOS ONE
---

## [Editor Report · Acceptance letter]

8 Jan 2025

PONE-D-24-37096R1 

PLOS ONE

Dear Dr. Alsetohy, 

I'm pleased to inform you that your manuscript has been deemed suitable for publication in PLOS ONE. Congratulations! Your manuscript is now being handed over to our production team.

Kind regards, 

on behalf of

Dr. Firomsa Bekele 

Academic Editor

PLOS ONE